# A Comprehensive Review of Channel Modeling for Land Mobile Satellite Communications

**Mauro Tropea *** and **Floriano De Rango**

Dimes Department, University of Calabria, Via P. Bucci 39/c, 87036 Rende, Italy; derango@dimes.unical.it
* Correspondence: mtropea@dimes.unical.it; Tel.: +39-0984-494-786

**Abstract:** As demands on the network continue to grow, it is increasingly important to upgrade the existing infrastructure in order to offer higher bandwidth and service level guarantees to users. Next generation networks (NGNs) represent a fully IP-based architecture that is able to support different technologies. In this context, the satellite networks are considered a fundamental part for future hybrid architectures. In this scenario, knowing satellite channel propagation characteristics in order to be able to design a communication system to respond to new user needs is of fundamental importance. Many papers in the literature show channel models in different satellite scenarios both for fixed and mobile applications; however, to the best of our knowledge, nobody presents an overview on different satellite models based on Markov chains. This paper wants to present a comprehensive review of the most recent developments in satellite channel communications related to mobile services and, in particular, for the land mobile satellite systems. The work presents all different types of Markov models, from single-state to multi-state models, that have been proposed in the literature from the early 1980s.

**Keywords:** land mobile satellite (LMS); satellite channel model; Markov model; mobile satellite communications





## 1. Introduction

Satellite platforms have been and continue to be a fundamental piece of telecommunications networks due to their ability to cover large geographical areas with their footprints and to their native broadcast nature, despite their development costs. In the context of next generation networks (NGNs), satellite systems represent a fundamental architectural component composing the overall future hybrid/multi-layer architecture. These networks are also able to satisfy the constant need of high bandwidth by new applications offering new typologies of services for fixed and mobile users, guaranteeing, with their appliance, the quality of service (QoS) requirement through IP mechanisms such as Multi-Protocol Label Switching (MPLS), integrated services, and differentiated services. Some of these QoS architectures involve pre-booking resources before reaching the required constraints, while others mark packets with priority and send them to the network without reservation [1,2].

The satellite services can be classified in two main categories on the basis of the user and application typologies that they try to service, namely, fixed satellite service (FSS) and mobile satellite service (MSS), that can use geosynchronous (GSO) and non-geosynchronous (NGSO) satellites, respectively. In the context of a mobile environment, a key role is reserved for the land mobile satellite (LMS) systems: a MSS system in which mobile Earth stations are located on land (Figure 1) [3]. Many studies of research on satellite platforms deal with the use of these networks together with other technologies such as high-altitude platforms (HAPs), unmanned aerial vehicles (UAVs) and terrestrial networks [4]. Some of these studies analyze QoS policies for unicast and multicast routing in multi-layered architectures, such as in [5], where the channel modeling can represent an important aspect to analyze. Other studies analyze the channel behaviour and how it affects switching policies between different network layers in hierarchical architectures, such as in [6].

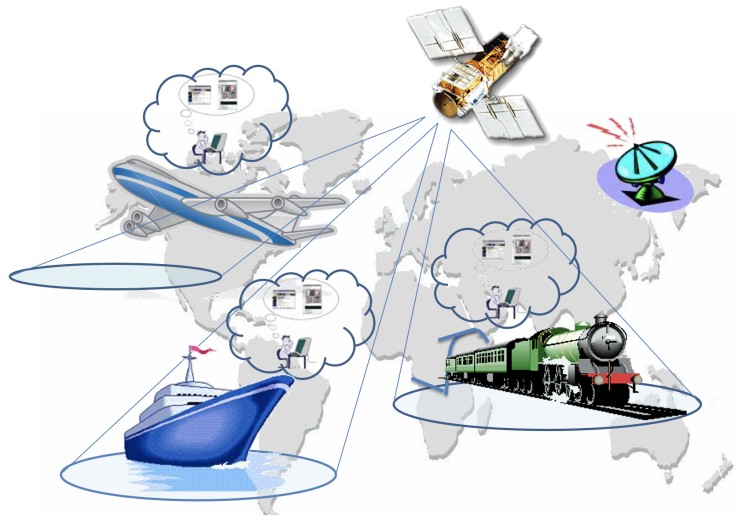

**Figure 1.** Mobile satellite system.

The great success of the satellite platforms are due mainly to some characteristics related to large bandwidth guaranteed by the use of high-frequency bands that permit the support of: small antenna sizes, making satellite services more pervasive and affordable for millions of commercial and residential end-users; larger system capacity, which is guaranteed by the use of smaller beams thanks to high frequencies, allowing the performance of a cellular-type mechanism, known as frequency reuse, with the capacity of serving more users in the covered area; ubiquitous access, thanks to broadcast nature of satellites and their capacity to reach those areas where other types of telecommunication services are unavailable.

Thus, the important satellite characteristics of providing an excellent signal quality in a wide coverage propagation are extremely useful in scenarios of natural disasters and make satellite communication very attractive. Moreover, in the context of NGN networks, the use of satellite communication together with 6G wireless technology is able to guarantee high data throughput and channel capacity, making these networks able to improve the QoS provided to mobile users [7]. In this context, the design of an opportune channel model that is able to model and reproduce high-frequency satellite communication characteristics represents a key factor for the scientific communities that deal with satellite systems [8].

The main issue typical of these network typologies is the severe signal degradation that is due mainly to atmospheric effects and territory orography. These negative effects are much more evident when working with high frequencies with respect to the use of lower ones. Therefore, negative meteorological conditions such as rain, ice, clouds, and, in general, gas absorption, give rise to a series of phenomena whose study is of fundamental importance in order to be able to deploy satellite networks with optimal performance.

In this scenario, being conscious of the propagation characteristics of the satellite channel, so as to be able to design a communication system for responding to the new user needs, is of fundamental importance. Thus, adequate knowledge of propagation phenomena is necessary for the performance assessment of these systems. This paper wants to present a comprehensive overview of the most recent developments in satellite channel communications related to mobile services and, in particular, for land mobile satellite (LMS) systems in urban, suburban, and rural environments by presenting all different types of Markov models, from single-state to multi-state models, proposed in the literature from the early 1980s.

The organization of the paper is as follows: Section 2 presents a brief review of the literature on LMS channel models. Section 3 describes the satellite characteristic fundamentals. Section 4 describes the propagation characteristics and phenomena of LMS communication. The LMS channel models proposed in the literature are provided and

described in Section 5. The paper ends with the future research direction and the concluding remarks provided in Sections 6 and 7, respectively.

## 2. Related Work

Starting from the early 1980s, a great number of works has been published in the literature regarding the study of channel models for LMS scenarios. For communication on LMS links, an adequate understanding of the different phenomena and impairments that affect satellite signal propagation in scenarios where mobile terminals are moving around is necessary. For what concerns the signal propagation, different phenomena such as reflection, scattering, diffraction, and multipath phenomena are to take into account in order to model a precise and real satellite channel model. In order to consider the correct statistical formulation, a classification of the communication environment is normally in urban, suburban, and rural scenarios. The main difference between fixed (FMS) and mobile satellite systems (MSS) is that the elevation angle of MSS is much larger, and this affects significantly the QoS received by mobile terminals.

Some of the most important publications in the literature regard studies of about forty years ago. One of the first studies on land mobile satellite communication regards an experiment conducted in order to determine an additional path loss on a free-space loss for LMS communications [9]. A statistical model for scenarios of land mobile satellites is provided by Loo in [10]. In this paper, the author provides a model for land mobile satellite communications in a LoS for most of the time and assumes that a lognormal distribution governs the LoS component under attenuation due to foliage (shadowing) and Rayleigh distribution for the multipath component. During the years since, a lot of papers have been proposed on satellite channel models and the common line of all these works is the use of a Markov chain approach that is based on a different number of states, from a two-state model to a multi-state model, that tries to capture the dynamic of the signal and propagation environment. Additionally, in the last years, some researchers have published articles about channel models for satellite communications; for example, in [11], the authors present a three-dimensional channel model at Q-band frequencies modeled by a Markov chain approach, performing experiments on the campus of Heriot-Watt University of Edinburgh. A very recent study [7], showing the synergy between satellite communication and 6G technology, proposes a novel atmosphere data-driven channel model based on artificial neural networks for Q-band frequencies that suffer from different propagation impairments due to high frequencies. In [12], an S-band satellite channel model is proposed in order to deal with the fading characteristic of satellite communication that affects satellite performance. Their simulation experiments, through actual channel measurements, verify the the accuracy of the proposed model. In [13], for the new satellite platform named *Cubesat*, a numerical tool to explore typical irradiation scenarios for CubeSat missions by combining state-of-the-art models is presented. The exploration of these new platforms is of great interest for the scientific communities in the field of wireless communications, and the aspects that concern the channel model are also to be investigated.

In the next sections, after presenting the main satellite characteristics and phenomena, the main LMS channel models proposed in the literature are described.

Our current research work's main motivation originates from the importance of studying land mobile satellite channel models in order to design an adequate channel model so as to guarantee optimal satellite performance in an overall network scenario. This research study aims to provide a brief and comprehensive review of the main satellite channel models in land scenarios in order to give to the researchers a survey document that comprises the main land mobile satellite channels proposed in the last four decades. This work is not meant to be an exhaustive work on satellite channel models for land environments, but it does provide a survey that shows how to deal with land mobile satellite channels and an extensive bibliography covering the last years of research. Then, the main contribution and the novelty of this work is the overview of the main satellite channel models for land scenarios, grouping a series of literature works from the early years of the 1980s.

## 3. Satellite Characteristic Fundamentals

Satellite telecommunications are a form of radio frequency telecommunications through satellite radio links between ground-based transceiver stations and artificial satellites in orbit around the Earth. These systems, made possible by the birth and development of launch technologies (rockets) starting in the second half of the twentieth century, often represent the only solution that is applicable in the absence of terrestrial infrastructures, or when such infrastructures are difficult to locate, and with an overall cost that is lower than the realization of equivalent terrestrial communication systems. In the following, a brief description of the main satellite characteristics are provided.

### 3.1. Satellite Transponder Technologies

The structures/appliances installed on the satellite that guarantee the consolidation and retransmission of the satellite signal are called transponders, as they transpose the reception channel (called uplink) with the transmission channel (called downlink), shifting the used frequencies. The device sends a signal in response to the received one. This system allows for transmitting a downlink channel, providing a wide range of services. There are two types of satellite transponders [14]:

- Transparent, also known as Bent Pipe (BP): the signals that reach the satellite through the radio channels can be sent back to Earth without undergoing any changes. As shown in Figure 2a, the main component of this satellite system is represented by the frequency converter block connected in input to a low-noise amplifier (LNA) receiver, and in output to a high-power amplifier (HPA) element. The signal is received by a receiving antenna and forwarded by the transmitting one after being handled by the three above blocks without changing its characteristics [15]. The Carrier-to-Noise (C/N) ratio is the main link parameter, and it represents the difference in decibels (dB) between the acceptable carrier signal strength and the unwanted noise power at the receiver;

- Regenerative, also known as On-Board Processing (OBP): using digital signals, this type of transponder is able to receive information from the ground and to re-transmit only after having processed the signals. In particular, the signals sent by users (uplink) that arrive at the satellite with distortions or noise will undergo down-conversion, demodulation, de-multiplexing, and reconstruction processes before they are retransmitted to the ground terminal (downlink) after performing modulation, multiplexing, and up-conversion processes [14]. The main OBP satellite blocks are depicted in Figure 2b, and they contribute to improving the overall satellite performance. Generally, for this typology of satellite platforms, the bit error rate (BER) parameter is considered in terms of bit error probability.

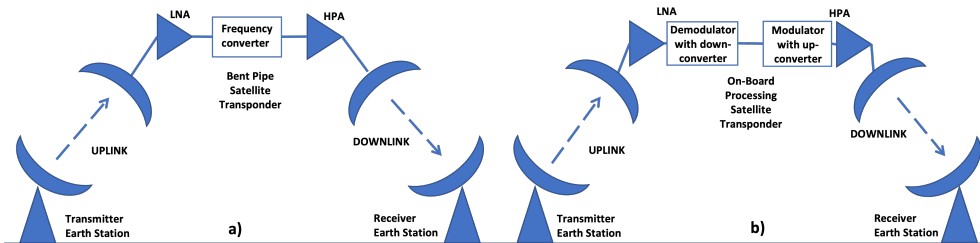

**Figure 2.** (**a**) Bent pipe transponder satellite; (**b**) on-board processor transponder satellite.

The satellite network is composed of a control station, called a network control center (NCC), that has the task of managing satellites, allowing the transmission. The users inside the coverage area can exploit satellite services available for both fixed and mobile users with or without the use of a gateway or a ground station that is able to manage aggregated traffic flows. Clearly, the mobile user terminals are small devices that are limited in power with a radio link that suffers from the variations in the space of the mobile-satellite user devices. User devices and gateways compose the ground segment, whereas the space

segment consists of satellite objects that orbit around the Earth and ground stations used for monitoring and controlling the satellites [14,15].

### 3.2. Satellite Orbit Classification

The main satellite classifications regard the orbits and altitudes in which they are placed (see Table 1); they can classified as [1]:

- Geostationary Earth orbit (GEO): these satellites are placed over the equator with the same angular speed of the Earth. The main characteristic of these satellites is that they appear in the sky as fixed points, so they are able to service the same constant area, also called a footprint (approximately 43% of the Earth's surface). They have an altitude of 35.786 km and the same rotation and direction of the Earth, with a propagation delay of 250–280 milliseconds (ms). GEO satellites are more suited for fixed communications, but recently, they have also been used for providing services to mobile users;
- Non-geostationary Earth orbit (NGEO): belonging to this category are satellites that, due to Van Allen radiation belts, are placed in three different orbits and altitudes. They are: low Earth orbit (LEO), at a height between 500 and 1500 km of altitude and with an end-to-end propagation delay of about 20–25 ms; medium Earth orbit (MEO), at a height between 7000 and 25,000 km of altitude and with a delay of about 110–130 ms; and highly elliptical orbit (HEO), at a height between 400 and 50,000 km. LEO and MEO satellite orbits have the advantages of being closer to the Earth and having lower latency, but have also the disadvantage of needing several satellites, normally called a constellation, in order to cover the entire Earth's surface, introducing handover procedures in the communication towards the users.

**Table 1.** Satellite Orbit Classification.

| Platform | Altitude | Orbit |
|---|---|---|
| Low Earth orbit (LEO) | 500–1500 km | circular |
| Medium Earth orbit (MEO) | 7000–25,000 km | circular |
| Geostationary Earth orbit (LEO) | 35,786 km | fixed |
| Highly elliptical orbit (HEO) | 400–50,000 km | elliptical |

### 3.3. Satellite Link Multiple-Access Techniques

In order to exploit adequately the precious satellite resource, an opportune multiple-access technique has to be used in communications. The used technique allows users to share the satellite channel, avoiding collisions when accessing the bandwidth resource. Generally, in satellite communications, four types of multiple-access techniques are used, namely:

- Frequency-division multiple access (FDMA);
- Time-division multiple access (TDMA);
- Code-division multiple access (CDMA);
- Multi-frequency-TDMA (MF-TDMA), a hybrid solution that exploits the FDMA and TDMA characteristics.

Moreover, LMS satellites are able to provide their services in a widespread range of spectrum bands [16], such as: L-band 1–2 GHz, S-band 2–4 GHz, C-band 3.4–6.725 GHz, Ku-band 10.7–14.8 GHz, Ka-band 17.3–21.2 GHz, 27.0–31.0 GHz, and Q/V-bands 37.5–43.5 GHz, 47.2–50.2 GHz and 50.4–51.4 GHz, and other bands as well; see Table 2.

**Table 2.** Satellite spectrum bands.

| Band | Frequency Range |
|------|-----------------|
| P | 0.2–1 GHz |
| L | 1–2 GHz |
| S | 2–4 GHz |
| C | 4–8 GHz |
| X | 8–12 GHz |
| Ku | 12–18 GHz |
| K | 18–26 GHz |
| Ka | 26–40 GHz |
| Q | 33–50 GHz |
| V | 40–75 GHz |
| W | 75–110 GHz |

*3.4. MSS Systems*

MSS systems constitute an important solution to providing communication services to mobile users in different conditions and scenarios, such as in under-populated regions, in emergency areas, and on planes, trains, or ships; see Figure 1. The main characteristics that are possible to exploit for these types of systems are represented by robustness communication, wide area coverage, and broadcast/multicast capabilities. However, these systems can be in the condition of non-line-of-sight (NLoS) propagation caused by the presence of obstacles in the path between a satellite and mobile terminals [17].

The first MSS system was realized by the International Maritime Satellite Organization (Inmarsat) in 1982, when it started to offer a global voice, data, and telex service. This system was designed for maritime mobile satellite communication applications, but the land mobile market was identified, after various studies and analyses, as the most important element [15].

The MSS systems are classified in three different categories [18]:

- Land mobile satellite (LMS) service;
- Maritime mobile satellite service;
- Aeronautical mobile satellite service.

## 4. Propagation Characteristics and Phenomena of LMS Communication

The satellite system object of this study is represented by a LMS system where it is possible to individuate two types of channels:

- Fixed channel: between NCC or gateway and satellite in the sky;
- Mobile channel: between mobile terminal and satellite.

Both channels have distinct features that must be considered in the system design process. The mobile channel is the most crucial of the two links, since transmitter power, receiver gain, and satellite visibility are limited compared to the permanent link. The mobile terminal works in a dynamic and frequently hostile environment with continuously changing propagation characteristics. Then, in this scenario, it is fundamental to be aware of the propagation phenomena because, for high frequencies (typically over 10 GHz), atmospheric and geographical conditions can severely affect satellite communication.

The characteristics of the satellite channel are different for different types of satellites, which can be geostationary or orbit on elliptical orbits at different heights. The fundamental parameter to characterize the satellite channel is the elevation angle under which the satellite is seen; in fact, with the satellite very high on the horizon, it is quite unlikely that the signal will be blocked by obstacles on the Earth's surface. When the orbit is low, or when the satellite is not geostationary, there is channel variability due to the fact that the satellite is seen moving on different elevation angles. In addition to what has just been said, the satellite channel is often also time-varying, due both to the fact that the Earth station and the satellite can move, and that atmospheric conditions can change [19].

In this section, the satellite propagation characteristics and propagation phenomena will be provided in order to show the main issues to be considered in a satellite system, other than propagation due to a direct path [20,21]. Moreover, in order to make the mathematical expressions in the section easier to follow, we provide a table listing the used mathematical symbols; see Table 3.

**Table 3.** Symbols used in Propagation Characteristics and Phenomena of LMS Communication section.

| Symbol | Description |
|---|---|
| $P_r$ | Receiver power |
| $P_t$ | Transmitted power |
| $d$ | Distance |
| $G_t$ | Transmitted antenna gain |
| $G_r$ | Receiver antenna gain |
| $\lambda$ | Wavelength |
| $p(a)$ | Probability density function (PDF) |
| $a$ | Received signal level |
| $\sigma^2$ | Received average power for multipath |
| $I_0$ | Bessel function of order zero |
| $K$ | Rice factor |
| $\frac{v^2}{2}$ | Direct component average power |
| $\mu$ | Mean of shadowed receiver signal component |
| $\sigma^2$ | Variance of shadowed receiver signal component |
| $(A_p)_u$ | Atmosphere corrective factor for uplink channel |
| $(A_p)_d$ | Atmosphere corrective factor for downlink channel |
| $\Lambda$ | Faraday rotation |
| $f$ | Frequency |
| $f_d$ | Doppler frequency |
| $s$ | Mobile terminal speed |

*4.1. Direct Wave*

The direct wave is the direct ray that arrives at the receiver site via a line-of-sight (LoS) path without reflection. Other than attenuation due to free space, an additional path loss, as a function of the elevation angle ($\theta$) of the station with respect to the ground, can be considered. Free space attenuation causes the received signal power to decrease with the distance between the transmitter and receiver. The receiver power at a distance $d$ between the transmitter and receiver is expressed by the following formula, known as the Friis or free space loss (FSL) equation:

$$P_r(d) = P_t G_t G_r \cdot \frac{\lambda^2}{(4\pi d)^2},$$
(1)

where $P_t$ is the transmitted power, $G_t$ and $G_r$ are the antenna gain of the transmitter and receiver, respectively, and $\lambda$ is the wavelength expressed in meters. For frequencies up to about 10 GHz, the propagation is not affected by rain.

The optimal conditions are when the transmission of the LoS component is not obstructed by obstacles. The LoS component is unaffected by shadowing, and the entire receiving signal is made up of the LoS (dominant) component plus a large number of independently fading multipath components that follow the Rice distribution [22], whose probability density function (PDF) is represented by:

$$p_{Rice}(a) = \frac{a}{\sigma^2} exp\left[-\frac{a^2 + v^2}{2\sigma^2}\right] I_0\left(\frac{av}{\sigma^2}\right),$$
(2)

where $a$ is the received signal level, $\sigma^2$ represents the average power received caused by multipath, and $I_0$ represents the zero-order Bessel function. The Rice factor, indicated with $K$, is defined as the ratio between the average power of the direct component ($v^2/2$) and that due to the multipath ($\sigma^2$):

$$K = \frac{v^2}{2\sigma^2}. \tag{3}$$

Some works, such as [23], propose the use of the Nakagami–Rice distribution for the LoS condition whose PDF is:

$$p_{Nakagami-Rice}(a) = \frac{v}{\sigma^2} exp^{-(v^2+a^2)/2\sigma^2} I_0\left(\frac{va}{\sigma^2}\right). \tag{4}$$

*4.2. Multipath Fading*

Small-scale variability is the one that best characterizes fading and that takes the countless multiple paths present in urban environments into account, which can produce additional attenuations in amplitude and the phase of a signal, typically over a short time period. This phenomenon is often caused by obstruction due to foliage or construction, resulting in diffraction, reflection, and dispersion. It is also known as multipath fading, and it is a typical problem of mobile radio communications studied and analyzed with characteristics typical of a random process. The Rayleigh distribution is used to characterize multipath fading, and the PDF of the signal envelope is given as:

$$p_{Rayleigh}(a) = \frac{a}{\sigma^2} exp\left[-\frac{a^2}{2\sigma^2}\right], \tag{5}$$

where $a$ is the received signal level, $\sigma^2$ represents the average power received, caused by the multipath.

*4.3. Shadowing Fading*

When the signal suffers from obstructions in the propagation path, its level decreases, and a shadowing effect occurrs because the LoS component is absorbed or dispersed. It is also referred to as large-scale fading. The attenuation of the direct path can be caused by different obstacles, such as roadside trees, buildings, hills, or mountains. The most dominant effect in this scenario is the shadowing that determines fading. This effect is strongly affected by elevation angles and by the frequency used in the system. The higher the frequency, the greater the effect due to obstructions such as trees. The degree of the signal attenuation shadowing effect, which is commonly characterized by a lognormal distribution, is characterized by the following PDF:

$$p_{Lognormal}(a) = \frac{1}{a\sqrt{2\pi}\sigma} exp\left[-\frac{(lna - \mu)^2}{2\sigma^2}\right], \tag{6}$$

with $\mu$ and $\sigma^2$ being the mean and variance of the shadowed component of the received signal, respectively [24].

*4.4. Additional Path Loss*

The additional path loss, as can be seen in [21], decreases when increasing the elevation angle, and it is zero when the angle is equal to 90 degrees. It is very complex to take into account the different losses due to the different layers of the atmosphere, and a simple corrective factor is to be considered both for the uplink ($-(A_p)_u$) and the downlink ($-(A_p)_d$) channels. A further loss is considered for the climatic conditions ($A_r$). Thus, the following equation is considered for the received power $p_r$:

$$\begin{aligned} P_r = \\ P_t + G_t + G_r + 147.6 - 20log(f \cdot d) - A_p - A_r = \\ EIRP + G_r + 147.6 - 20log(f \cdot d) - A_p - A_r, \end{aligned} \tag{7}$$

where $EIRP$ is the equivalent isotropic radiated power, $147.6 - 20log(f \cdot d)$ represents the attenuation of FSL, and $A_p$ represents the loss due to atmosphere on the generic link (uplink or downlink). The term $d$ is a length in km: $d = 42{,}643.7 \cdot \sqrt{1 - 0.295577 \cdot (cos\phi \cdot cos\delta)}$,

where $\phi$ is the latitude and $\delta$ is the longitude of the ground station. Through these angles, it is possible to determine the elevation angle ($\theta$) from the horizon.

### 4.5. Faraday Rotation

The telecommunications satellites are placed in orbits above the ionosphere; therefore, the frequencies that can be used for connections with satellites must be sufficiently high to prevent them from being reflected by the ionosphere. That is, they must be higher than a few tens of MHz. When crossing the ionosphere, phenomena such as Faraday's rotation and ionospheric scintillation must also be considered [24].

Faraday's rotation is the rotation of the polarization axis of a wave caused by some characteristics of the atmosphere. Its formula in radians can be expressed as follows:

$$\Lambda = \frac{2.36}{f^2} \int_d B_L N dl, \tag{8}$$

where $B_L$ is the component of the Earth's magnetic field along the direct path $d$, $N$ is the electron density (electrons per $m^3$), and $f$ is the frequency in Hz. It has been verified that, in the worst conditions for the worst 1% of the year, the loss introduced is 3 dB. Loss can be eliminated by using circular polarizations.

### 4.6. Ionospheric Scintillation

Ionospheric scintillation is produced by irregularities in the electron density in the ionized layers of the ionosphere. It represents a cloud of electrons with a density that is highly different from the ionosphere layers. These non-homogeneities create reflections and diffusions of the radio waves, but the contributions of this diffusion rapidly decrease as the frequency increases, so their contribution can be ignored. Non-homogenous ionized layers cause scattered reflection of radio waves in the L-band, resulting in fluctuations in the amplitude and phase of the received signal. The scattered signals decrease rapidly as the frequency increases. The use of high frequencies is also motivated by the need to use small-sized antennas with very high gain that compensate for the considerable attenuation due to the large distances involved, taking into account the limited powers that are available on the satellites [24].

### 4.7. Tropospheric Effects

The tropospheric effects are important for the signal propagation and they are more significant at higher frequencies. The signal is affected by different phenomena due to hydrometers, such as rain or clouds, and atmospheric gases, such as oxygen, water vapor, etc. [24].

### 4.8. Doppler Shift

The motion of the mobile terminal causes a change in the properties of the environment, giving the received signal a theoretically non-stationary statistic; however, in practice, the channel can be described as almost stationary, as its characteristics can be considered slowly variable (elevation-angle types of obstacles, surface irregularities). The carrier undergoes a Doppler shift due to the movement of the terminal; this shift can be calculated based on the formula:

$$f_d = \frac{s}{c} \cdot f_c \cdot cos\theta, \tag{9}$$

where $f_c$ is the carrier frequency, $s$ is the mobile terminal speed, $c$ is the speed of the light, and $\theta$ is the elevation angle of the direct component [24].

As a result of this shift, the phase of the generic contribution of the indirect component in reception will have a phase shift of $2\pi f_d t$. The bandwidth of this component for mobile terminal speeds around 100 km/h is about 200 Hz.

*4.9. Reflection*

The reflection is the phenomenon in which the signal is reflected from the ground in the direction of the satellite, and it depends on the elevation angle of the satellite as seen from the mobile satellite terminal. The magnitude of the ground-reflected wave is proportional to the terrain roughness factor, and its effect can be reduced by terminal antenna directivity [19,24].

## 5. LMS Channel Models

LMS platforms are gaining a lot of interest in the current generation of wireless systems and are expected to gain even more interest in the next generations due to the feasible services and their ability to serve many users over a large area at a low cost. LMS systems are becoming increasingly important for different types of applications, including navigation, communications, and broadcasting. The satellite channel models are based on statistical formulations that have the task of taking into account different propagation characteristics, as shown in Section 4.

The model which is of interest for authors is for a LMS channel in a different land environment, such as urban, suburban, rural road, or highway environments where a LoS signal component is available at the receiver for a time more or less greater than that for multipath components.

A complex model must be used to represent the LMS channel [25]. This model tries to characterize the different conditions to which the signal received on the ground can be subjected, unlike other channel models in which the effect of the environment on the received power is incorporated in a single distribution with appropriate statistical parameters that take into account the type of fading to which the signal is subject. In practice, what is used to characterize the channel between a satellite and a terrestrial mobile station is a different model for the various visibility conditions to which the signal (state) can be subjected; subsequently, the individual processes are combined into a single global process based on the transition probabilities of the states. The basic idea is that a random sequence can be generated by different 'sources', each of which can be described with a very specific statistical model representing a different 'state' of the signal. The link between the various states can be expressed through a Markov chain with as many states as there are different statistical models; therefore, the switching process between the various channel models is described with the Markov chain. The use of a Markov channel model is an approach used for different types of wireless networks [26], such as underwater acoustic channel communications [27,28], ultra wideband (UWB) networks [29], and vehicle ad hoc networks [30].

The description of the channel is, therefore, extremely important, although very difficult: in fact, communications via satellite to a terrestrial mobile terminal have the drawback of undergoing considerable variations in the received power because the signal, before reaching the terrestrial mobile station, must pass through the various layers of the atmosphere—because it is subject both to fading due to multiple paths, and to absorption (obscuration) due to obstacles (shadowing).

Fading occurs when the signal is not only received by a direct path (LoS), but also has reflected components. This means that, for certain periods of the link, the received signal may present a power lower than that foreseen by the theoretical link budget, due to the various components that are added to the receiver in an inconsistent way. On the other hand, shadowing occurs when the propagation path between the satellite and the terrestrial mobile station is obstructed by natural elements (mountains, trees, and so on) or by built structures.

A statistical model for the envelope of the signal received in an LMS channel is useful for predicting the performance of a communication system with various modulation schemes.

In the following, a classification of satellite channel models on the basis of the number states of a Markov chain is presented in order to provide a comprehensive review of the

possible satellite channel model to be used and to know what represents the adding of a new state in the model. Table 4 gives the main mathematical symbols used in the mathematical formulas provided in the following.

**Table 4.** Symbols used in the LMS channel model section.

| Symbol | Description |
|---|---|
| $a$ | Signal amplitude |
| $\phi_0, \phi$ | Phases uniformly distributed in $(0, 2\pi)$ |
| $b_0$ | Average scattered power due to multipath |
| $p(a)$ | Probability density function (PDF) |
| $\sqrt{d_0}$ or $\sigma$ | Standard deviation of the signal |
| $\mu$ or $m$ | Mean of the signal |
| $p_i j$ | Transition probabilities for Markov chain |
| $G$ | Good state |
| $B$ | Bad state |
| $R$ | Transmission rate |
| $v$ | Mobile terminal speed |
| $\hat{\pi}$ | Stationary state vector for Markov process |
| $P$ | Transition matrix for Markov process |
| $M_{r,x}$ | Mean multiple power |
| $m$ | Mean of the signal |
| $\sigma$ | Standard deviation of the signal |
| $x$ | Received voltage |
| $f_1, f_2, f_3$ | Cumulative distribution function |
| $P_1, P_2, P_3$ | Occurrence probability of the three states |
| $d$ | Distance |
| $erf$ | Error function |

As the works in the literature prove, the main used and proposed satellite channel model is represented by the three-state model that captures propagation channel variations while considering a LoS state and two other states concerning low-shadowing and high-shadowing conditions. Table 5 summarizes the channel models used in the specific reference papers, and makes it possible to see the main characteristics of each model.

**Table 5.** LMS Channel models, summarized.

| Author/References | Year | Model | Statistic | Frequency band | Environment | Remarks |
|---|---|---|---|---|---|---|
| Loo [10]/[31] | 1985/1998 | Single-State | Lognormal/Rayleigh | UHF-, L-band/UHF-, L-, S-, Ka-band | Rural | - |
| Corazza [32]/[33] | 1994/1994 | Single-State | Rice/Lognormal | L-band/L-band | Urban/Rural/Suburban | - |
| Xie [34] | 2000 | Single-State | Rice/Lognormal | Ku-band | Urban/Suburban/Rural | - |
| Lutz [35]/[36] | 1991/1996 | Two-State | Rice/Lognormal/Rayleigh | L-band/- | City/Highway | - |
| Aboderin [37] | 2015 | Two-State | Lutz model | L-band | - | Mobile terminal is transiting within two different propagation environments |
| Rougerie [38] | 2016 | Two-State | Loo model | Ku-, Ka-band | Urban/Rural/Suburban/Highway | Apply Loo model to Ka-/Ku-band |
| Akinniyi [39] | 2017 | Two-State | Lutz model | L-band | Different Environments | Satellite diversity approach was employed in addition to the 2-state Markov chain |
| Karasawa [40] | 1997 | Three-State | Rice/Lognormal/Rayleigh | - | Urban/Suburban | Satellite diversity effect assuming that the area is illuminated simultaneously by at least two satellites moving in LEO for urban and suburban environments |
| Fontan [41] | 2001 | Three-State | Loo model | L-, S-, Ka-band | Different Environments | - |
| Braten [23] | 2002 | Three-State | Nakagami–Rice/Rayleigh | L-band | Heavily wooded/suburban | - |
| Milojevic [42] | 2009 | Three-State | Loo model | - | Urban/Suburban/Rural | Dynamic higher-order Markov-state model for joint processes that depends on the current state duration for both single- and multiple-satellite broadcasting systems |
| Liu [43] | 2016 | Three-State | Loo and Corazza model | L-band | Urban/Suburban/Rural | - |
| Bai [11] | 2019 | Three-State | - | Q-band | Urban/Suburban/Rural | It contains three parts: FSPL model, a modified shadowing model based on a first-order Markov-chain process, and a small-scale fading based on a 3D geometry-based stochastic model (GBSM). |
| Fontan [44] | 1998 | Three-State | Loo model | S-band | Open/Suburban/Rural/Urban | - |
| Scalise [45] | 2008 | Three-State | Rice/Lognormal/Rayleigh | Ku-band | Urban/Rural/Suburban/Highway | - |
| Iglesias [46] | 2012 | Four-State | Nakagami–Rice/Loo/Rayleigh | L-, S-band | Urban/Suburban/Heavily wooded/Lightly wooded/Rural | - |
| Lutz [36] | 1996 | Four-State | Rice/Lognormal/Rayleigh | - | City/Highway | - |
| Ming [47] | 2008 | Five-State | Lutz model | - | Different Environments | - |
| Dongya [48] | 2005 | Six-State | Rice/Lognormal/Rayleigh | L-band | Different Environments | - |
| Ming [49] | 2008 | Six-State | Lutz model | - | Different Environments | - |
| Shen [50] | 2005 | Six-State | Rice/Lognormal/Rayleigh | L-, S-band | Different Environments | - |
| Babich [51] | 2000 | Multi-State | Rice/Lognormal/Rayleigh | narrow-band | - | Quantized fading |
| Hsieh [52] | 2001 | Multi-State | Rice/Lognormal/Rayleigh | - | Urban/Suburban/Wooded/Rural | Different fade-level states |
| Zhang [53] | 1999 | Multi-State | Rayleigh | - | - | Rayleigh Fading Channels, where each state corresponds to different channel quality indicated by BER |
| Tropea [54,55] | 2013 | Multi-State | - | - | - | Idea is not for analyzing a single packet, but for fixing an observation window and evaluating the QD of the link, computing the packet error rate (PER) associated to the specific window. |
| Guo [56] | 2014 | Multi-State | - | Ka-band | - | The channel of the meteorological factor principal component is obtained by using the principal component analysis method, and the fuzzy clustering analysis method is introduced into channel classification. |

Moreover, Table 6 shows a comparison between the three main environments analyzed in the different literature works: urban, suburban, and rural, from the point of view of statistics types used in the Markov models. The described Markov models are based on different numbers of states, which are used for better representing the propagation channel characteristics and for better capturing the evolution of the satellite channel through the use of specific communication parameters, such as BER, fading states, and others.

**Table 6.** Urban, suburban, and rural environment statistic comparison.

| Environment / Markov Model | Urban | Suburban | Rural |
|---|---|---|---|
| Single-State | Rice/Lognormal | Rice/Lognormal | Rice/Lognormal Lognormal/Rayleigh |
| Two-State | Rice/Lognormal/Rayleigh Lognormal/Rayleigh | Rice/Lognormal/Rayleigh Lognormal/Rayleigh | Rice/Lognormal/Rayleigh Lognormal/Rayleigh |
| Three-State | Lognormal/Rayleigh Rice/Lognormal/Rayleigh | Lognormal/Rayleigh Rice/Lognormal/Rayleigh Nakagami–Rice/Rayleigh | Rice/Lognormal/Rayleigh Rice/Lognormal/Rayleigh |
| Four-State | Nakagami–Rice/Lognormal/Rayleigh | Nakagami–Rice/Lognormal/Rayleigh | Nakagami–Rice/Lognormal/Rayleigh |
| Five-State | Rice/Lognormal/Rayleigh | Rice/Lognormal/Rayleigh | Rice/Lognormal/Rayleigh |
| Six-State | Rice/Lognormal/Rayleigh | Rice/Lognormal/Rayleigh | Rice/Lognormal/Rayleigh |
| Multi-State | Rice/Lognormal/Rayleigh | Rice/Lognormal/Rayleigh | Rice/Lognormal/Rayleigh |

| PDF expression / PDF type | Mathematical Expression | | |
|---|---|---|---|
| Rice | $p_{Rice}(a) = \frac{a}{\sigma^2} exp\left[-\frac{a^2+v^2}{2\sigma^2}\right] I_0\left(\frac{av}{\sigma^2}\right)$ | | |
| Lognormal | $p_{Lognormal}(a) = \frac{1}{a\sqrt{2\pi}\sigma} exp\left[-\frac{(lna-\mu)^2}{2\sigma^2}\right]$ | | |
| Rayleigh | $p_{Rayleigh}(a) = \frac{a}{\sigma^2} exp\left[-\frac{a^2}{2\sigma^2}\right]$ | | |
| Nakagami–Rice | $p_{Nakagami-Rice}(a) = \frac{v}{\sigma^2} exp^{-(v^2+a^2)/2\sigma^2} I_0\left(\frac{va}{\sigma^2}\right)$ | | |

### 5.1. Single-State Channel Model

A statistical model for LMS systems is represented by the Loo model [10,57]. It assumes that the amplitude of the LoS component under foliage attenuation (shadowing) is lognormally distributed and that the received multipath interference has a Rayleigh distribution. In the following, a brief mathematical description of the model is provided. This model previews that the signal is modeled using the sum of the lognormal and Rayleigh random variables with independent phases, as shown:

$$a \cdot exp(j\theta) = l \cdot exp(j\phi_0) + r \cdot exp(j\phi), \tag{10}$$

where $a$ represents the signal amplitude, $\phi_0$ and $\phi$ are the phases uniformly distributed in $(0, 2\pi)$, and $l$ and $r$ represent the lognormal and the Rayleigh distribution, respectively. The PDF of $a$ is given by:

$$p_{Loo}(a) = \frac{a}{b_0} \int_0^\infty exp\left[\frac{-(a^2+l^2)}{2b_0}\right] I_0\left(\frac{al}{b_0}\right) p(z) dz, \tag{11}$$

where $b_0$ represents the average scattered power due to the multipath and $I_0(\cdot)$ is the modified zero-order Bessel function. Considering $a$ as the lognormal for large values and distributed as Rayleigh for small values, the $p(a)$ can be represented by:

$$p_{Loo}(a) = \begin{cases} \frac{1}{a\sqrt{2\pi d_0}} exp\left[\frac{-(lna-\mu)^2}{2d_0}\right], & for\ a \gg \sqrt{b_0} \\ \frac{a}{b_0} exp\left[\frac{-a^2}{2b_0}\right], & for\ a \ll \sqrt{b_0}, \end{cases} \tag{12}$$

with $\sqrt{d_0}$ and $\mu$ being the standard deviation and the mean, respectively.

Moreover, in order to allow integration over $(0, R)$ instead of $(R, \infty)$, it is possible to use the following relation:

$$p(a > R) = \int_R^\infty p_{Loo}(a) da = 1 - \int_0^R p_{Loo}(a) da \tag{13}$$

It is worth noting that, if the attenuation due to shadowing (lognormal distribution) is maintained constant, the PDF in Equation (12) just provides a Rician distribution. Experiments in rural regions with elevation angles of up to 30 degrees have been used to validate this model [31].

Other important contribution to this type of channel model exist in the literature, one of which is represented by the Corazza model [33]. In this paper, the authors introduce a statistical channel model that uses the combination of two important distributions, Rice and lognormal, to model the effects of the LoS and shadowing components. The principle provided in the work can be suitable for all types of environments, such as rural, urban, and suburban, thanks to a set of parameters that can be tuned opportunely. Moreover, the proposed model is suitable for non-geostationary satellite channels such as LEO or MEO orbits. The mathematical formulation for the proposed model is provided, also based on other authors' work [32], in which the channel model for both cellular and satellite communications is described for the statistical characterization under different propagation conditions. Another important work is represented in [34], where the authors take into account outages due to obstruction of the LoS path caused by interference from multipath radio waves for providing a statistical model suitable for mobile communications more generally, with respect to the well-known previous statistical models.

### 5.2. Two-State Markov Channel Model

Figure 3 shows a two-state Markov model for a LMS channel [35–39,57]. The LMS channel can be represented by a Gilbert–Elliott (GE) model, in which it is possible to distinguish between time intervals with high received power corresponding to a so-called *good* channel state, and time intervals with low power level corresponding to a so-called *bad* channel state. In particular, the good channel state represents the condition of LoS in the satellite communication characterized by a low packet loss probability, whereas the bad channel state is responsible for representing areas in which there are obstacles between the satellite and user terminal, characterized by an extremely high probability of packet loss [58]. The *good* channel state, called *G* and numbered as 1, corresponds to LoS situations; the *bad* channel state, called *B* and numbered 2, represents situations where the signal is blocked by an obstacle in the propagation path. The test courses were carefully selected to represent different types of environments (city, suburbs, rural roads, highways), and to comprise a mixture of cruising directions.

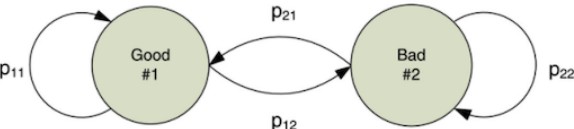

**Figure 3.** Two-state Markov channel model.

In this GE (two-state Markov model), the assumption is made that the system remains in a state for a certain number of seconds before deciding to make a transition. The transition can happen towards the other state or the same (current) state. The transition probabilities are labeled with $p_{11}$, $p_{12}$, $p_{22}$, and $p_{21}$, where $p_{nm}$ represents the probability of transitioning from state $n$ to state $m$. Only two of these transition probabilities are independent because of the relationships $p_{11} + p_{12} = 1$ and $p_{22} + p_{21} = 1$. Clearly, more complex satellite channel models have been proposed in the literature, which will be provided in the next section; however, the GE model represents the simplest yet reasonably realistic model of satellite fading.

Considering a mobile user with speed $v$, and indicating with $B$ the *bad* state and with $G$ the *good* state, the average extensions (in meters) of the shadowed and non-shadowed areas, $Area_B$ and $Area_G$, correspond to average time intervals $Int_B$ and $Int_G$, in which the channel remains in *bad* or *good* condition, respectively. Considering a transmission rate $R$,

the mean state durations normalized to the symbol duration and the transition probabilities $p_{21} = G$ and $p_{12} = B$ result in:

$$\begin{cases} Int_B = \frac{1}{p_{21}} = \frac{1}{G} = \frac{R}{v} \cdot Area_B \\ Int_G = \frac{1}{p_{12}} = \frac{1}{B} = \frac{R}{v} \cdot Area_G \end{cases} \tag{14}$$

The more representative model which first used the two-state model was that of Lutz [35]. This model uses a Rician distribution for describing the good channel behaviour representative of the LoS component, and a mixed Rayleigh and lognormal distribution incorporated in the Suzuki model for describing the NLoS condition representative of the bad channel state. The PDF of the Suzuki model [59], describing the bad channel in which obstacles can block the direct signal component, and of the Lutz model are as follows:

$$p_{Suzuki}(a) = \int_0^{+\infty} p_{Rayleigh}(a|\sigma^2) p_{Lognormal}(\sigma^2) d\sigma, \tag{15}$$

$$p_{Lutz}(a) = C \cdot p_{Rice}(a) + (1 - C) \cdot p_{Suzuki}(a), \tag{16}$$

where C and $(1 - C)$ represent the percentage of shadowing and unshadowing, respectively.

### 5.3. Three-State Markov Channel Model

Figure 4 shows a three-state Markov model for LMS communications. This model proposes considering the receiving signals as composed of a LoS component and multipath components. Three types of fading channels are defined, state #1: LoS condition; state #2: moderate shadowing condition; state #3: deep shadowing condition. The change among channel states is a relatively slow process [41]. In this model, each state represents a specific channel state and, in particular, $S1, S2, S3$ denote the respective channel states, and $P_{ij}$ is the probability that the Markov process will go from state #$i$ to state #$j$. The switching among each state is described by a transition matrix $P$, which is represented in Figure 7.

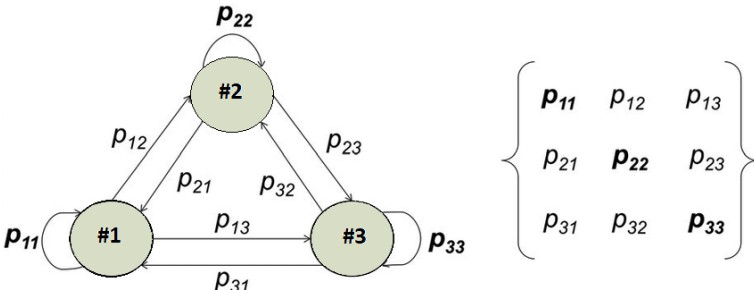

**Figure 4.** Three-state Markov channel model.

Based on the properties of the Markov process, a stationary state vector $\hat{\pi}$ is calculated as defined, $\hat{\pi} \cdot P = P$; then:

$$\hat{\pi} = (\pi_1 \pi_2 \pi_3). \tag{17}$$

Many papers in the literature propose the use of a three-state Markov channel model, taking into account the LoS, moderate shadowing, and deep shadowing or blackage [11,23,40–43]. In [44] the authors present a LMS propagation channel model/simulator based on a three-state Markov model plus the Loo distribution. They provide the results of an experimental campaign performed at different elevation angles and environments. Another work proposing this type of model is [41]. In this work, the authors present a model that is capable of describing both narrow- and wide-band conditions, and it is also suitable for links in both GEO and NGEO satellites. Moreover, they provide model parameters extracted from a comprehensive experiment in different environments and elevation angles at the L-, S-, and Ka-bands. A measurement campaign for the LMS systems in the Ku-band is presented

in [45], where the authors provide a series of experiments achieving data and comparing two-state and three-state channel models, taking into account the main propagation phenomena in a satellite environment.

### 5.4. Four-State Markov Channel Model

A first four-state Markov channel model was proposed by Lutz in [36] as a model for two correlated land mobile satellite channels, as shown in Figure 5. However, the four-state model for a single channel is proposed in [46]. A semi-Markov model for the low-elevation satellite–Earth propagation channel is presented in [46], where semi-Markov means no $P_{ii}$ arches. The model provided by the authors describes long-term variations of the received signal by a chain of four distinct states. This represents a novel satellite channel model that takes into account the classical LoS, shadowing and blockage states, and a new state in order to model all those periods of time when the receiver is unable to carry out the acquisition and tracking of the satellite.

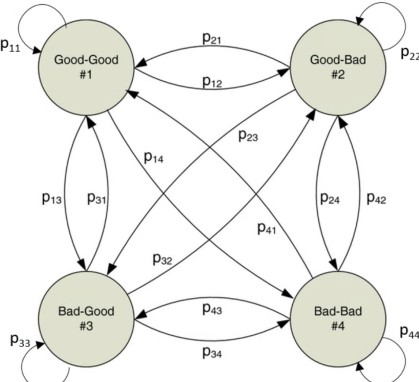

**Figure 5.** Four-state Markov channel model.

In particular, the authors propose a propagation model in which the LoS signal and multipath (state #1), due to surrounding elements, is modeled by a Nakagami–Rice distribution with a cumulative distribution function (CDF) so defined [60]:

$$f_1(x \leq x_0) = \int_0^{x_0} \frac{2x}{M_{r,1}} exp\left(-\frac{1+x^2}{M_{r,1}}\right) I_0\left(\frac{2x}{M_{r,1}}\right) dx, \tag{18}$$

where $I_0$ is a modified Bessel function of the first kind of zero order, $M_{r,1} = 2\sigma^2$ is the mean multipath power described in [60], and $x$ is the received voltage.

The shadowing (state #2) also consists of LoS and multipath components, but in this case, the LoS component is attenuated by trees and/or small obstacles. A good representation for this state is the model presented by Loo in [10], where the LoS signal follows a lognormal distribution and the reflected component follows a Rayleigh distribution:

$$f_2(x \leq x_0) = \frac{6.390}{\sigma M_{r,2}} \int_0^{x_0} x \int_\epsilon^\infty \frac{1}{z} exp\left(-\frac{[20log(z)-m]^2}{2\sigma^2} - \frac{x^2+z^2}{M_{r,2}}\right) I_0\left(\frac{2xz}{M_{r,2}}\right) dzdx, \tag{19}$$

where $\epsilon$ is a very small value ($\epsilon = 0.001$ is suggested), $m$ and $\sigma$ are the mean and standard deviation of the signal fading in decibels for the direct wave component, and $M_{r,2}$ is the mean multipath power.

The third state (state #3) is represented by blockage, that is, the state in which the LoS component is blocked by large obstacles, so no LoS contribution, only the multipath, is present. The fading distribution for this state is described in the ITU-R model [58] by a Rayleigh distribution with CDF:

$$f_3(x \le x_0) = 1 - exp\left(-\frac{x_0^2}{M_{r,3}}\right), \tag{20}$$

where $M_{r,3}$ is the mean multipath power of the diffuse reflections.

Taking into account the above formulations for the three cases, the cumulative distribution function of the received signal envelope can be expressed as:

$$CDF(x \le x_0) = P_1 \cdot f_1 + P_2 \cdot f_2 + P_3 \cdot f_3, \tag{21}$$

where $P_1$, $P_2$, and $P_3$ are the occurrence probability of the three states that fulfill the equation $P_1 + P_2 + P_3 = 1$.

The authors have added a new state to the three-state model for modeling those periods when the fade caused by a large obstacle leads the signal to values lower than the sensitivity of the receiver, and neither direct nor multipath components can be recovered, so the receiver is not able to track the satellite. This new state, indicated with #4, will have an initial probability of $P_4$. The #4 state #4 scope is mainly included to provide better knowledge of the duration of fades deeper than the sensitivity of the receiver that cause the interruption of the channel. The CDF of the state #4 is expressed by:

$$CDF(x \le d) = \frac{1}{2}\left[1 + erf\left(\frac{ln(d) - m}{\sigma\sqrt{2}}\right)\right], \tag{22}$$

where $m$ and $\sigma$ are the mean and standard deviation of the logarithm of the distance $d$, and *erf* is the *error function* [60].

Another work in the literature [61] proposes the use of a four-state Markov model for LMS satellite channel for a dual-satellite MIMO communications able to improve the BER performance under the same two considered satellites' signal-to-noise ratio condition.

### 5.5. Five-State Markov Channel Model

In [47], a new channel model with a five-state Markov chain for LMS systems is presented. The particularity of the proposed approach is to merge the two-state model with the three-state one in order to form the new five-state model. The authors specify three different areas for the five different states: low-shadowing area (LSA), moderate-shadowing area (MSA) and high-shadowing area (HSA). For each area, the authors define different link conditions; in the LSA, they define two states: LoS and low-shadowing condition; in MSA, they define only the moderate-shadowing condition, and in HSA they define the high- and entire-shadowing conditions. From Figure 6, it is possible to note that the state #1 or state #2 does not pass to the state #4 or state #5, and there are no passes from state #4 and state #5 to state #1 and state #2, so the transition matrix $P$ can be represented as depicted in Figure 9.

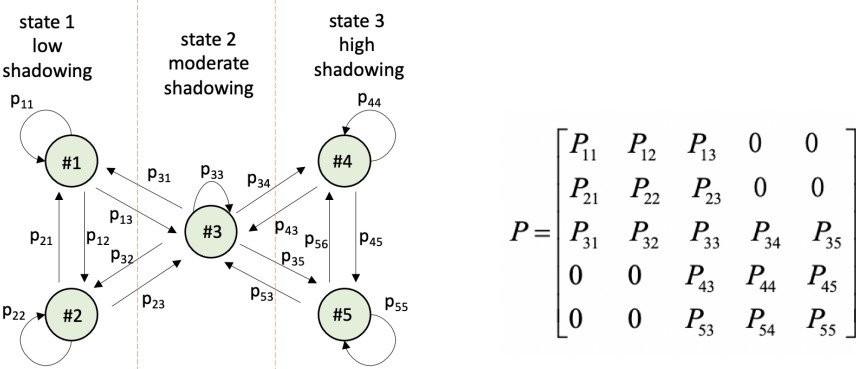

**Figure 6.** Five-state Markov channel model .

## 5.6. Six-State Markov Channel Model

A novel channel model with a six-state Markov chain for a LMS communication system is proposed in [48,49]. As depicted in Figure 7, the authors propose a six-state Markov chain inside two different shadowing areas, named state #1: low-shadowing and state #2: high-shadowing, that forms a two-state GE model. When the received amplitudes are in the state #1: low-shadowing or in the state #2: high-shadowing for a certain time, the model allows them to pass between the three sub-states. Moreover, in this case, as previously shown for the five-state Markov chain, the transition matrix *P* has a specific value due to the impossibility of passing between specific states, as can be seen in Figure 7. The authors, in fact, introduce the concept of transition states: the state #1 or state #2 cannot transfer themselves to the state #5 and state #6 directly; they must pass through the transition states (state #3 and state #4; see Figure 7). It is reasonable to assume that the state #1 is LoS and the state #6 is blockage [50].

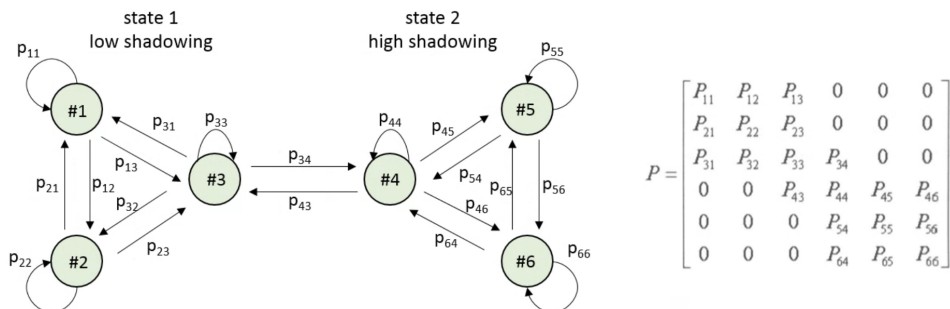

**Figure 7.** Six-state Markov channel model.

## 5.7. Multi-State Markov Channel Model

Other works in the literature show a satellite channel model approach based on a multi-state Markov chain [51]. In [52], the authors propose a channel model with more states but that is capable of limiting transition matrix size to save processing time and memory requirements for predicting fading properties and error probability performance for LMS. They have built their model based on typical measured data, using slow fading modeled as a lognormal distribution and fast fading modeled as a Rayleigh distribution. The proposed two-level, multi-state Markov model gives satisfactory predictions of first- and second-order statistics of propagation properties for satellite communications [62]. Other works exist in the literature that propose satellite channel models based on a Markov chain with a number *n* of states Figure 8. For example, in [26,53], the authors develop and analyze a methodology to partition the received signal-to-noise ratio (SNR) into a finite number of states according to the time duration of each state. Then, their proposed finite-state Markov chain is composed of a number of states where each of them corresponds to different channel quality represented by the fading associated with the channel. In addition, in [54,55], the authors describe in detail a satellite channel for a Digital Video Broadcasting–Return Channel Satellite (DVB-RCS) mobile system. They present a high-level satellite channel model based on the Markov approach. The proposal introduces the concept of windowed observations, and the idea in the paper is that of analyzing more packets in a certain window in order to evaluate the link degradation, computing an analysis of the packet error rate (PER) associated with a specific observation window. The analysis conducted by the authors does not consider only a two-state (good and bad) model but, introducing the concept of quality degree (QD), they analyze and compute the PER belonging to each window, discretizing the PER values in order to generate the correct number of states for the considered Markov model; see Figure 9. Therefore, the number of additional states of the classical GE model for augmenting the accuracy of the model depends on the specific operative conditions and, thus, also on the noise power level, satellite type, climatic conditions, and error correction rate.

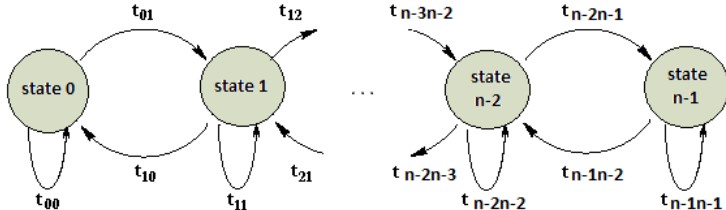

**Figure 8.** Multi-state Markov channel model.

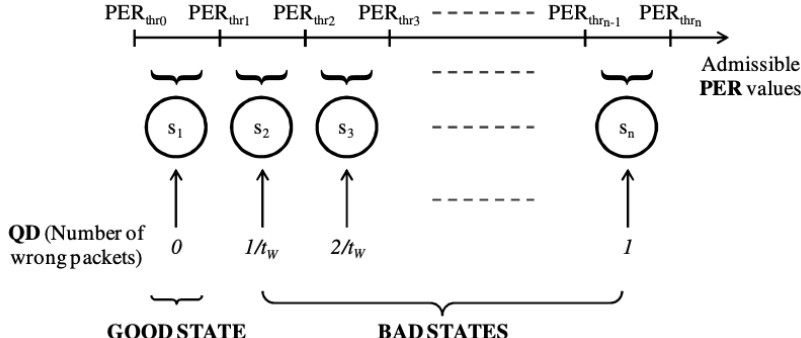

**Figure 9.** PER discretization for multi-state Markov model.

In [56], the authors propose an analysis of the Ka-band satellite channel propagation characteristic and how meteorological factors affect it. They introduce, firstly, the principal component analysis method, and then a fuzzy clustering analysis in the channel study, providing a multi-state Markov model that proved its accuracy throughout simulation experiments.

## 6. Future Research Direction

Satellite propagation channels have significant importance in optimizing the coverage, reliability, and capacity performance of satellite communication. Many works in the literature provide different models to be used for designing an opportune satellite channel model; however, despite this, many research issues remain open. Measurement campaigns are beneficial for the formulation of effective satellite channel models, evaluating the performance of their communication systems, and network planning. However, this type of studies are strongly dependent on regional environments. In fact, many works in the literature base their studies on different types of scenarios, such as urban, suburban, rural open areas, and so on, in order to capture the correct issues of the satellite propagation phenomena. Many other studies are based on stochastic approach, and propose satellite channel models based on the Markov chain approach in order to be able to capture adequately the different propagation characteristics of the satellite channel.

In the last years, with the development of 6G technology and the new hierarchical architectures in which different wireless channels are contained, the studies on satellite channel models at different and always higher frequencies is of fundamental importance in order to guarantee the best performance for the users' applications [8,63,64]. Moreover, the birth of the new Cubesat technology represents another important motivation for continuing to analyze, study, and propose new channel models that are able to always boost network performance [65]. As shown in [66], the new frontier of satellite networks has progressed toward optical communications. In this work, the authors have investigated a robust power-allocation strategy for the downlink, taking into account the effects of the atmospheric impairments. Moreover, they have evaluated their proposal through simulative campaigns using experimental channel measurements from the ESA project on an optical satellite called ARTEMIS. Other studies [67,68] show the use of non-orthogonal multiple access (NOMA) into an integrated satellite–terrestrial network in order to exploit the advantage of this technique for being able to provide the required QoS to the satellite

users and to guarantee better performance for the overall system. A key role in both satellite platforms and multi-layer architectures is played by scheduling schemes that determine an important improvement on the sytem performance, as shown in [69]. Also, massive multiple-input multiple-output (MIMO) is a promising new technology that is largely used and proposed in satellite scenarios, thanks to its capacity to enhance spectral efficiency, exploiting the slow-varying statistical channel state information (CSI) at the transmitter [70].

## 7. Conclusions

The fast growth of telecommunications and the NGN networks based on a hybrid/ multi-layer architecture make the study of satellite platforms of fundamental importance for the development of a network architecture that is able to satisfy the requests of new users' applications. This work provides, after a brief description of the propagation problems of LMS communication links and the statistical resolutions, an overview on LMS channel models provided in the scientific literature. Adequate knowledge of propagation impairments and channel models is required for designing and evaluating the performance of the advanced technologies used to establish reliable communication links in LMS communication systems. The main objective is to highlight the effects and the related propagation patterns that must be considered for LMS communication links in order to accurately estimate the propagation issues. The performance of LMS communication systems depends on several factors, including operating frequency, elevation angles, geographic location, climate, etc. Thus, the main task of this paper is to collect the main literature works about satellite channel models, starting from the 1980s and progressing up to the present day, providing a comprehensive review about the stochastic model approach to the channel design. Different methodologies, such as more realistic physical-statistical channel models, may be used to determine the impact of these elements on LMS communications, but they need extensive and complex simulations, whereas stochastic approaches are easy, more desirable, and need less computing work. Furthermore, due to the different natures of propagation environments, a Markov-based approach can represent an important tool for modeling satellite channel characteristics, guaranteeing the QoS and performance required by users. As shown in the paper, different Markov models using a different number of states have been proposed in order to try to capture, in a better way, the propagation characteristics of satellite links. Through this review, it was discovered that the most-used Markov model is that which proposes three different states, as is also suggested by the ITU recommendation in [58], where the states represent deep-shadow, intermediate-shadow, and good-state link conditions.

**Author Contributions:** Conceptualization, F.D.R. and M.T.; methodology, F.D.R.; formal analysis, M.T.; writing—review and editing, F.D.R. and M.T., supervision, F.D.R. All authors have read and agreed to the published version of the manuscript.

**Funding:** This research received no external funding.

**Institutional Review Board Statement:** Not applicable.

**Informed Consent Statement:** Not applicable.

**Data Availability Statement:** Not applicable.

**Conflicts of Interest:** The authors declare no conflict of interest.

## Abbreviations

The following abbreviations are used in this manuscript:

| | |
|---|---|
| BP | Bent Pipe |
| C/N | Carrier-to-Noise ratio |
| CDF | Cumulative Distribution Function |

| | |
|---|---|
| CDMA | Code Division Multiple Access |
| CSI | Channel State Information |
| DVB-RCS | Digital Video Broadcasting–Return Channel Satellite |
| FDMA | Frequency Division Multiple Access |
| FSS | Fixed Satellite System |
| GE | Gilbert–Elliott |
| GEO | Geostationary Earth Orbit |
| HEO | Highly Elliptical Orbit |
| HPA | High-Power Amplifier |
| LEO | Low Earth Orbit |
| LMS | Land Mobile Satellite |
| LNA | Low-Noise Amplifier |
| LoS | Line-of-Sight |
| MEO | Medium Earth Orbit |
| MIMO | Multiple-Input Multiple-Output |
| MF-TDMA | Multi-Frequency Time-Division Multiple Access |
| MPLS | Multi-Protocol Label Switching |
| MSS | Mobile Satellite System |
| NCC | Network Control Centre |
| NGEO | Non-Geostationary Earth Orbit |
| NGN | Next Generation Network |
| NLoS | Non-Line-of-Sight |
| NOMA | Non-Orthogonal Multiple Access |
| OBP | On-Board Processing |
| PDF | Probability Density Function |
| PER | Packet Error Rate |
| QD | Quality Degree |
| QoS | Quality of Service |
| SNR | Signal-to-Noise Ratio |
| TDMA | Time-Division Multiple Access |

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
