# Peer review of "A Comprehensive Review of Channel Modeling for Land Mobile Satellite Communications"

_electronics, doi:10.3390/electronics11050820_

Round 1
Reviewer 1 Report
This paper reviews the channel modeling for the land mobile satellite communication. This paper is well presented with interesting contents. The reviewer has the following concerns:
(1) Various channel models are reviewed in this work, which are presented in a rather straightforward way. The motivation of researchers proposing each model is not clear enough. The usage scenario of each channel model should be explained more clearly.
(2) Since there are many types of channel modeling, which one is the most commonly used, and which one is rarely used. It would be better to give some comparisons between different models. Furthermore, it would be better to give a table to summarize which channel model is used by which reference papers.
(3) The introduction section may be further improved to clarify the importance of satellite communicaiton channel modeling for the 6G networks or for the practical usage.
(4) The current research points and the future research directions of channel modeling for the satellite communications are not clear. It would be better to give a more comprehensive discussion of the channel modeling, from the past, current, to the future research.
Reviewer 2 Report
- Introduction is not clear and needs more information in context to why authors have chosen the topic for the review. I would recommend authors to revise the introduction section again and add relevant literature listing why the specific topic is chosen.
- Figures 2 and 3 should be redrawn with good quality image and text readable. Hence, I would like to recommend authors to revise and redraw figures 2 and 3.
- Conclusion section is missing information in context why this research work has been done, what is the significance of the work and how it would benefit all. Hence, authors should relevant information in conclusion.
Reviewer 3 Report
the Article "A Comprehensive Review of Channel Modeling for Land Mobile Satellite Communications"
It has an interesting theme and if better explained some points can contribute significantly to the magazine and readers.
Therefore, it is imperative that the authors describe a section related to related works, comparing the work performed in relation to the researched literature. Describing criteria and parameters used, make clear and evident the novelty in relation to your work.
I suggest that among the articles compared, at least half are from the last 3 years (2020, 2021 and 2022).
These recently published works may be useful for the work developed, they have related themes and topics.
https://doi.org/10.3390/s21092914
https://doi.org/10.3390/en13246691
They are from the same publisher and have, in addition to related topics, the structure and definitions necessary for a journal article.
It is necessary to better describe the tests and results obtained, I cannot understand which and how they were performed, much less reproduce in other scenarios.
In the conclusions, it is not clear which research problems were solved, the novelties that this article presents and especially the academic contributions. I strongly suggest describing what future work will be.
I suggest entering a list of mathematical symbols.
Strongly review the bibliographic references, several are not complete and do not have a DOI or ISSN.
Round 2
Reviewer 1 Report
The reviewer's concerns have been addressed satisfactorily. The reviewer has no further comments. Thanks.
Author Response
We thank the reviewer for the review and the suggestion provided.
Reviewer 2 Report
- Abstract is not clear. Authors should add few lines or more relevant information in context to significance of work done in the article, in comparison to exiting literature.
- For the better understanding of readers, Authors should add a table and compare the scenarios of 'urban, suburban and rural environment' in context to the importance and significance in the article and with the existing literature.
Reviewer 3 Report
Changes were made, for the final version to revise writing and text formatting.
Author Response

(The authors gave the same response as above.)
